# Bioisosteric Replacement as a Tool in Anti-HIV Drug Design

**DOI:** 10.3390/ph13030036

**Published:** 2020-02-28

**Authors:** Alexej Dick, Simon Cocklin

**Affiliations:** Department of Biochemistry & Molecular Biology, Drexel University College of Medicine, Rooms 10307, 10309, and 10315, 245 North 15th Street, Philadelphia, PA 19102, USA; ad3474@drexel.edu

**Keywords:** bioisosteres, HIV-1, antiviral, computer-aided drug design, envelope, reverse transcriptase, protease, integrase, tat, Vif

## Abstract

Bioisosteric replacement is a powerful tool for modulating the drug-like properties, toxicity, and chemical space of experimental therapeutics. In this review, we focus on selected cases where bioisosteric replacement and scaffold hopping have been used in the development of new anti-HIV-1 therapeutics. Moreover, we cover field-based, computational methodologies for bioisosteric replacement, using studies from our group as an example. It is our hope that this review will serve to highlight the utility and potential of bioisosteric replacement in the continuing search for new and improved anti-HIV drugs.

## 1. Introduction

The design and development of a lead compound into a drug is a laborious and often costly process, with most candidates failing due to metabolism and pharmacokinetics issues rather than potency. Bioisosteric replacement is a strategy used by medicinal chemists to address these limitations while still retaining the potency/efficacy of the initial lead compound. The use of bioisosteres and the introduction of structural changes to the lead compound allows the chemist to alter the compound’s size, shape, electronic distribution, polarizability, dipole, polarity, lipophilicity, and pKa, while still retaining potent target engagement. Therefore, the bioisosteric approach can be used for the rational modification of a lead compound towards a more attractive therapeutic agent with improved potency, selectivity, altered physical, metabolic, toxicological properties with the bonus of generating novel intellectual property (IP). The objective of the present review is to provide a broad understanding of the principle of bioisosteric replacement underlined with cases from recent applications in anti-HIV drug design and optimization. Specific examples for lead compound development targeting a variety of HIV-1 targets, including the envelope (Env), reverse transcriptase (RT), protease, integrase (IN), Tat, and Vif, will be presented. Studies from our group that document the application of bioisosteric replacement to the ongoing progressive alteration of a dominant piperazine chemotype in the HIV-1 entry inhibitor field, and which suffers from bioavailability and breadth problems, are also covered. It is our hope that this review will serve to highlight the utility and potential of bioisosteric replacement in the continuing search for new and improved anti-HIV drugs.

## 2. Principle of Bioisosterism and Historical Background

The term isosterism was first introduced by Irving Langmuir in 1919 during his studies on similarities of physicochemical properties of atoms, groups and molecules [1]. He described compounds or groups of atoms with the same number of atoms and electrons, such as N_2_ and CO, N_2_O and CO_2_, or N_3_^-^ and NCO^-^ as isosteres, and, based on these similarities of the arrangement of electrons, he defined 21 groups of isosteres. H. G. Grimm further developed this definition in the early 1920s. This early hypothesis of bioisosterism describes the ability of certain chemical groups to mimic other chemical groups [2,3]. Accordingly, the addition of a hydride to an atom gives to the resulting pseudoatom the properties of the atom with the next highest atomic number (Table 1) [4]. 

Each vertical column represents an isostere, according to Grimm. In 1932, Hans Erlenmeyer extended Grimm’s definition of isosteres as atoms, ions, and molecules in which the peripheral layers of electrons (valence electrons) are considered as identical (Table 2). 

Based on its application in biological systems, Harris Friedman introduced the term “bioisostere” in 1950 that included all atoms and molecules which fit the broadest definition for isosteres and have similar biological activity, either agonistic or antagonistic [5]. Today, the even more broadened definition of bioisosteres introduced by Alfred Burger in the early 1990s is in use. Accordingly, bioisosteres are “Compounds or groups that possess near-equal molecular shapes and volumes, approximately the same distribution of electrons, and similar physical properties” [6]. 

## 3. Classical and Non-Classical Bioisosteres

In the 1970s, Alfred Bruger defined bioisosteres as either classical (atom number, number of valence electrons, and degree of unsaturation) or non-classical (similar pKa, electrostatic potentials, orbital occupation/HOMOs and LUMOs) [7]. Classical bioisosteres can be further subdivided into five classes: 1) monovalent atoms or groups (D and H; F and H; C and Si; Cl, Br, SH, and OH; NH and OH; RSH and ROH, –Cl, –PH_2_, –SH), 2) divalent atoms or groups (–CH_2_, –NH, –O, –S, –Se–, -COCH_2_-), 3) trivalent atoms or groups (–CH=, –N=, -P=, -As=), 4) tetravalent atoms or groups (>C<, >Si< and =C=, =N^+^=, =P^+^=), and 5) ring equivalents (-CH=CH-, -S- (e.g., benzene, thiophene), -CH=, -N= (e.g., benzene, pyridine), -O-, -S-, -CH_2_- (e.g., tetrahydrofuran, tetrahydrothiophene, cyclopentane) [4,8]. Non-classical bioisosteres are a more sophisticated mimicry of the emulated counterparts and they do not strictly obey the steric and electronic definition of classical isosteres. As non-classical bioisosteres can significantly differ in electronic distribution, physicochemical, steric and topological properties, they have found beneficial applications in drug discovery research. Non-classical bioisosteres are subdivided into two groups: 1) cyclic vs. non-cyclic and 2) exchangeable functional groups (Figure 1) [4]. 

In the following sections, we will highlight the recent, successful application of both classical and non-classical bioisosteres in the design and redesign and development of new anti-HIV agents. 

### 3.1. Recent Applications for Classical Bioisosteres in Anti-HIV Drug Design and Development

#### 3.1.1. Monovalent Bioisosteres 

##### Deuterium as a Hydrogen Isostere in HIV-1 Reverse Transcriptase Inhibitors

Deuterium (^2^H, heavy hydrogen), as one of two isotopes of hydrogen with an isotopic mass of 2.014 u, is the most common isostere of hydrogen. Both isotopes have small differences in their physicochemical properties beneficial for drug design. Deuterium provides lower lipophilicity, smaller molar volume, and a shorter (by 0.005Å) C–D bond as compared to the C–H bond. Deuterium incorporation can also increase the basicity of amines and decrease the acidity of phenols and carboxylic acids [8]. As a result, intermolecular interactions with the target can be affected. Covalent bonds, or adjacent bonds, in which the replaced deuterium is involved, can display altered cleavage properties [9]. Non-covalent interactions can be changed as well; however, the effect is usually modest. Nevertheless, if located adjacent to modification sites, measurable effects are observable, for example, in modulating the drug candidate’s pharmacokinetic properties such as its metabolism and toxicity. 

Retroviruses encode an enzyme known as reverse transcriptase (RT) to convert their viral single-stranded RNA into linear double-stranded DNA for integration into the host genome and subsequent transcription and replication. This process is essential for the viral replication cycle and the target of one of the most successful inhibitor class, the RT inhibitors (RTI) [10]. RTI can be subdivided into two major types, nucleoside (nucleotide) analog reverse-transcriptase inhibitors (N(t)RTIs) and non-nucleoside reverse transcriptase inhibitors (NNRTIs). 

Mutlib and colleagues showed that the widely used HIV-1 NNRTI efavirenz (1) produced renal tubular epithelial cell necrosis in rats (Figure 2) [11]. To understand the biochemical mechanisms of forming toxic, reactive intermediates and to characterize the safety of efavirenz, Mutlib et al. used a hydrogen-to-deuterium exchange approach to improve the metabolic stability. Efavirenz undergoes a complex metabolic transformation pathway that leads ultimately to the nephrotoxic glutathione conjugate (3). Deuteration at the cyclopropyl moiety significantly reduced the formation of the cyclopropylcarbinol intermediate (2) and the excretion of (3) in the urine of rats as quantified by LC/MS.

##### Silicon as a Carbon Isostere in HIV-1 Protease Inhibitors

Silicon has reawakened the interest of medicinal chemists over recent years as a bioisosteric replacement for carbon (so-called “silicon switching”). Carbon and silicon, despite both having a valency of 4, have the ability to form tetrahedral compounds with slightly different chemical properties. The most notable differences afforded by replacing C with Si are the increased covalent radius of Si (50%), the longer C–Si bond (20%), decreased electronegativity according to the Pauling scale of Si (1.74 compared to 2.50 of C) and the higher lipophilicity of Si derivatives [12,13]. These differences can result in modest shape and conformation differences of silanols, crucial for the protein–ligand interaction. Silicon, as compared to carbon, also displays a lower tendency to form stable π bonds, and the Si–Si σ bond (230 kJ mol^−1^) is weaker compared to the Si–O bond (368 kJ mol^−1^). Therefore, silicon appears as silicates and silica in nature. The ability of Silanols to form hydrogen bonds, combined with their acidic character as compared to carbinols, make them an attractive replacement for optimizing hydrogen bonds in protein–ligand interactions [14]. To date, no Si-related toxicity has been recorded, which makes Si an attractive candidate for drug discovery. Silicon modified derivatives can alter metabolic pathways or improve blood-brain barrier penetration due to its increased lipophilicity. Alterations can vary from simple alkyl replacement by a trialkylsilyl to the more sophisticated silandiols (Si(OH)_2_). 

Highly active anti-retroviral therapy (HAART) is a very effective treatment for AIDS, and protease inhibitors (PIs) play a crucial role in HAART. PIs block the proteolytic cleavage of viral precursor proteins into shorter active proteins essential for the viral replication cycle. Silinadiol represents a very important class of protease inhibitors, highlighting the potential of Si for this research area. For example, proteases hydrolyze the peptide amide bond with a tetrahedral diol intermediate that further reacts to the individual peptides (Figure 3). 

Mimicking this transition state is a powerful strategy for inhibitor design. Silicon, in fact, due to its preference to appear in sp3 hybridization over sp2, stabilizes the geminal transition state diol and prevents further hydrolysis into a silanone (Si=O) as pioneered by Sieburth [15]. The reduced electron-withdrawing character of Si disfavors the dehydration of the silanone moiety and stabilizes the transition state (Figure 4, right equilibrium) [8]. 

Using a high-pressure liquid chromatographic (HPLC) assay, to quantify the rate of HIV-1 protease inhibition based on the cleavage of a HIV-1 Gag peptide substrate, the silanediol (4) inhibited HIV-1 protease with a K_i_ of 2.7 nM and in a plaque assay with human peripheral blood mononuclear cells (PBMCs) an EC_90_ of 170 nM similar to the carbinol (5) (K_i_ = 0.37 nM, EC_90_ = 23 nM, Figure 3 green dashed box) was determined. 

Most of the HIV-1 protease inhibitors are accompanied by side effects in long-term treatment. HIV protease inhibitor-induced metabolic syndromes can include dyslipidemia, insulin resistance, and lipodystrophy/lipoatrophy, as well as cardiovascular and cerebrovascular diseases [16]. Therefore, safer and potentially promising protease inhibitor development is highly desirable. Silicon-based protease inhibitors provided a proof of principal for the use of this isostere and future safety evaluation studies [15,17].

#### 3.1.2. Divalent Bioisosteres

Divalent isosteres can be classified into two subgroups: (1) atoms that are involved in a double bond such as C=C, C=N, C=O, and C=S and (2) those divalent isosteres where substitution of different atoms results in the alteration of two single bonds such as in the series; C-C-C, C-NH-C, C-O-C, and C-S-C [4]. Both bioisosteric classes are popular and have been used in the past extensively.

##### Ether/Sulfone Substitution in HIV-1 Protease Inhibitor Design

The highly potent HIV-1 protease inhibitor saquinavir (IC_50_ = 0.23 nM in an in vitro HIV-1 Gag peptide substrate cleavage assay) suffers from poor bioavailability due to its peptidic nature and NH content. Absolute bioavailability in healthy volunteers receiving oral saquinavir 600 mg was approximately 4% [18]. The reason for its poor bioavailability is thought to be a combination of incomplete absorption and extensive first-pass metabolism. X-ray structures in complex with this inhibitor class showed extensive interactions with the protease backbone, in particular, an extensive hydrogen bond network was shown to be crucial for biological activity. However, as drug resistance emerged, Gosh and colleagues focused on the retention of a maximum number of contacts of inhibitor with the protein backbone to combat mutant enzymes [19].

A crucial asparagine moiety in saquinavir that contacts the backbone of Asp29 and Asp30 in HIV-1 protease (Figure 5A) was replaced with 3(R)-tetrahydrofuranylglycine to improve inhibitory potency (Figure 5B, IC_50_ = 0.05 nM in an in vitro HIV-1 Gag peptide substrate cleavage assay) [19,20].

To potentially improve bioavailability, Gosh and colleagues replaced this 3(R)-tetrahydrofuranylglycine moiety in amprenavir (K_i_ = 160 nM) with a cyclic sulfone group, in which the oxygen can accept H bonds from both Asp29 and Asp30 (Figure 5C) with retained potency (K_i_ = 1.4 nM). To retain the ability to address both asparagines, further optimization did lead to the design of a more lipophilic bicyclic ether as a sulfone isostere, within the highly potent protease inhibitor darunavir (Presista^®^, Janssen Therapeutics), which has an absolute bioavailability of 37% [21] (as compared to 4% by saquinavir) and was licensed in the USA in 2006 (Figure 5D) [19,20].

### 3.2. Non-Classical Bioisosteres in Anti-HIV-1 Drug Design and Development

The following section will describe the major class of non-classical bioisosteric replacements and in silico methods for bioisostere identification with recent applications for HIV-1 entry inhibitor design and development. Non-classical bioisosteres are all replacements that are not defined by the classical definition. They usually are and can be very structurally different with an altered number of atoms, electronics or other physicochemical properties but their replacement mimics the spatial arrangement of the original moiety such that biological activity is retained. As non-classical bioisosteres can be so different, they are most often employed in “scaffold hopping”, allowing the diversification of chemotypes for a given target. According to Patani et al., non-classical bioisosteres can be subdivided into the replacement of 1) cyclic groups by non-cyclic groups and 2) replacement of functional groups with a variety of chemical moieties to retain biological activity (see also Figure 1) [4]. The utility and use of non-classical bioisosteres have been greatly furthered by advances in novel computation representations of compounds. Computational exploration of bioisosteric replacements for scaffold hopping can facilitate the drug design and development process even for smaller companies or academic groups, who typically do not have access to the large screening facilities of big pharma. At the end of the following section, therefore, we describe the advantages of this approach with highlights from our research group for HIV-1 Env inhibitor design.

#### 3.2.1. Heterocyclic Bioisosterism

Heterocycles are important structural elements in drug design. Variation in size and shape can provide substitution over a wide range, while inherent electronic and physical properties are of importance in mediating drug–target interactions. If chosen carefully, they can modulate crucial properties in drug design such as H-bond donor (NH, OH, CH) or acceptor properties, electron-withdrawing or donating effects, and the potential to engage in π–π interactions [8]. Tautomerism offers additional opportunities to optimize the topographical presentation of substituents and drug–target interactions. The azole bioisosterism will be briefly described for the HIV-1 integrase inhibition.

Integration of the viral DNA into the host genome is a crucial step in the HIV-1 and other retroviruses life cycle and is performed by the virus-encoded integrase [22]. The catalytic process involves the coordination of Mg^2+^ by the HIV-1 integrase. Azoles, considered as amide surrogates, were extensively studied as HIV-1 integrase inhibitors [23,24,25,26,27]. In contrast to six-membered heterocycles, azoles are able to adopt a coplanar arrangement with additional metal-chelating elements presented by a series of pyrido [1,2-a]pyrimidine and 1,6-naphthyridine-based integrase inhibitors (Figure 6). Of particular importance is the nitrogen arrangement in the 1,2,4-oxadiazoles (Figure 6B,C). The nitrogens lone electron pair is crucial for the coordination of a second Mg^2+^ ion and activity against resistant viruses. In the pyrido [1,2-a]pyrimidine series, the thiazole (Figure 6A) was chosen for future optimization against clinically relevant resistant mutants.

#### 3.2.2. Bioisosterism of Functional Groups

Replacement of functional groups does not always result in the retention of biological activity. Nevertheless, a few examples will be highlighted in the following sections, with a focus on guanidine and amidine replacements. Guanidine and amidines are basic functionalities and protonated at physiological pH. This basicity and protonation can reduce membrane permeability, reducing active compound concentrations in target cells, and contributing to leads to poor bioavailability. Besides the replacement of the entire guanidine or amidine entity (pKa of 13–14), the adjacent CH_2_ group can be replaced with a carbonyl (pKa of 8) or an oxygen atom (pKa of 7–7.5) to reduce basicity [8].

##### Guanidine Bioisosteres

Upon HIV-1 entry, genomic RNA is reverse transcribed and integrated into the host cell chromosome. Subsequently, the hijacked host RNA polymerase II (RNAP II) initiates transcription of this proviral genome back into genomic RNA, and elongation is stimulated by the HIV-1 Tat protein [28]. This mechanism involves the binding of Tat to a cis-regulatory sequence in the viral 5′ Long Terminal Repeat (LTR) [29]. This stem-loop RNA element is termed TAR RNA [30]. This activation has implications for HIV replication and activation of latent virus reservoirs. The inhibition of this interaction represents a very attractive approach for AIDS treatment and cure.

In an RNA-targeted SAR study, Lee and colleagues discovered a novel peptidomimetic that blocks this interaction, by replacing the guanidine group with a squaryldiamine as a potent bioisostere (Figure 7). Although the new squaric acid diamide analog showed a 4-fold decrease in affinity (from a K_D_ of 1.8 uM to 7.7 uM), as measured using an in vitro competition assay based on fluorescence resonance energy transfer (FRET), it was the first bioisostere to mimic the guanidine group successfully [31].

##### Amide and Ester Bioisosterism

The amide bioisosterism is gaining popularity nowadays because of its implications for peptide chemistry and development of peptide mimetics. It is of particular interest for medicinal chemists to replace peptide bonds into chemically more stable and bioavailable entities. Esters, however, are usually replaced to increase metabolic stability, because esters are rapidly cleaved in vivo. Heterocyclic rings such as 1,2,3-oxadiazoles (6), 1,3,4-oxadiazoles (7), triazoles such as 1,2,4-triazoles (8), 2-isoxazoline (9), and imidazoline entities (10) are very popular for amide or ester replacements (Figure 8) [4]. In light of the diversity of amide and ester bioisosterism, only one example using heterocycles for the design of HIV-1 accessory protein viral infectivity factor (Vif) will be discussed here.

The HIV-1 Vif is an accessory protein and crucial for viral replication by antagonizing the host’s innate immunity [32,33]. Vif directly targets the human DNA-editing enzyme APOBEC3G (A3G), which catalyzes hypermutations in viral DNA as a defense mechanism [34]. HIV-1 Vif has no cellular homologs and is, therefore, a very attractive target for antiviral intervention. The RN-18 class of Vif inhibitors enhances A3G-dependent Vif degradation, increasing A3G incorporation into virions, and enhance cytidine deamination of the viral genome [35,36,37]. However, this class of inhibitors suffers from low potency and metabolic stability. Mohammed and colleagues successfully used bioisosteres of the central amide bond in RN-18 (Figure 9, pink moiety) such as 1,3,4-oxadiazole (11), 1,2,4-oxadiazole (12), 1,4-disubstituted-1,2,3-triazole (13), and 1,5-disubstituted-1,2,3-triazole (14) to develop new chemotypes of HIV-1 Vif inhibitors for further optimization [38].

## 4. In Silico Molecular Field-Based Scaffold Hopping for Non-Classical Bioisostere Identification

Scaffold hopping using bioisosteric replacement has always been an integral part of the conventional drug development process but has expanded in recent years due to the availability of facile computational methods. High throughput screening (HTS), which tests thousands or millions of compounds, is a mainstay in the pharmaceutical industry. However, by applying computationally driven bioisosteric replacement to such initial hits, chemotype diversification can be readily achieved, which in turn increases the probability of successfully carrying one of these chemotypes through to the clinic.

Of these computational methods of non-classical bioisostere identification, the use of molecular field points has gained increasing recognition over recent years and has been very successfully utilized by our group in our initial efforts to redesign a first in-class entry inhibitor to address non-optimal drug-like properties and bioavailability. Typical approaches that do not use molecular field points tend to view the bioisosteric fragment in a vacuum, taking only fragment comparison into account and not factoring in the effect of the replacement across the whole molecule. Using molecular fields to facilitate a comparison of the entire spatial physicochemical properties across the entire molecule, rather than the structure alone, can and has led to discoveries of non-intuitive bioisosteric replacements that have positively modulated drug-like properties while retaining target engagement [39,40,41,42,43]. However, molecular fields are continuous and vary with the conformation of the compound, and sampling at defined points in the three-dimensional grid can be computationally expensive (Figure 10A). Therefore, program collections such as Cresset (Cresset, Litlington, UK) use only extrema of these fields, the so-called field point, where intermolecular interactions most likely occur (Figure 10B).

This calculation and comparison of field points is based on the eXtended Electron Density Force Field Theory (XED) [44,45]. XED takes into account positive and negative electrostatic, Van der Waals, and hydrophobic field points. In contrast to most molecular mechanics force field methods that apply atom-centered partial charges, XED generates negative pseudo-orbitals surrounding the positive nuclei and distributes partial charges according to the orbital location. XED, therefore, mimics the electron distribution in a more accurate way by taking π-clouds and lone pairs into account, which play important roles in biological systems. In practice, Field Points of two or more compounds can be compared by sampling conformations to maximize the fields for optimal alignment, and in contrast to a structural alignment, new compounds with different structures but similar numbers and distribution of Field Points can be found. This so-called “templating” of molecules can also be used to generate a high-content pharmacophore for therapeutically interesting targets for which there is no known information regarding the small molecule binding site. The following sections will describe recent applications using such computational approaches from our lab for the design of novel HIV-1 entry inhibitors with improved potency, specificity, and ADME properties.

### Scaffold Hopping for Potency and ADME Improvement of HIV-1 Entry Leads

The inhibition of HIV-1 entry is an attractive, yet underexploited therapeutic approach for several reasons. First, much of the entry process occurs in water-soluble compartments easily accessible to drugs. Second, both viral and host cell components involved in HIV-1 entry have been identified and can be targeted. Third, because inhibition of virus entry prevents the host cell from becoming permanently infected, such inhibitors are potentially useful in many different modalities, including pre-exposure prophylactics, prophylactics, and microbicides [46].

In an effort to bring this class of inhibitor out of the realm of salvage therapy, we have applied both high-content, field-point pharmacophores to perform virtual screening, and iterative bioisosteric replacements, to identify novel leads for HIV-1 entry inhibitors. The most potent and broadly acting HIV-1 inhibitors to date are piperazine-based introduced by Bristol-Myers Squibb [47]. However, this class has low bioavailability, resulting from low solubility and poor dissolution. Bristol-Myers Squibb has partially overcome these limitations by adopting a prodrug approach with BMS-663068 that shows increased solubility in the gut. However, after removal of the phosphonooxymethyl-solubilizing group in the gut, the properties of the active compound, BMS-626529, dominate and, despite its potency, 1.2 g per day is required to achieve an effective plasma concentration [48]. Clearly, an entry inhibitor with more intrinsic drug-like properties would be preferable [49]. Therefore, we used field-based three-dimensional similarity virtual screening experiments using Blaze (Cresset, Litlington, UK) with a high-content field-based pharmacophore template derived from BMS-626529 and two of its predecessors, BMS-488043 and BMS-378806, to identify novel scaffolds that could function as entry inhibitors [50]. Field templating was implemented, as at the time of this study, no structural information regarding the binding site nor the bioactive conformation of the BMS compounds was available. FieldTemplater (Forge) allows the individual to generate a bioactive conformation hypothesis, and a field-point pattern pharmacophore, in the absence of structural information, as long as multiple, distinct compounds are known and are supposed to interact with the same site on the target. Figure 11 shows the field-point template generated and used in our HIV-1 entry inhibitor studies (Figure 11A), in addition to a comparison of the template to the experimentally determined bioactive conformation of BMS-626529 (Figure 11B) [51,52].

From this field-point pharmacophore screen using Blaze (Cresset UK), we identified hits with piperazine-based cores similar to the BMS chemotypes but with potencies in the low micromolar range, ranking from 13 to 153 µM using a HIV-1 YU-2 Env pseudotyped virus in a single-round infection assay (SRIA). Therefore, to obtain a truly novel core scaffold, we chose to conduct bioisosteric replacement (scaffold hopping) with Spark (Cresset, UK). The results of this indicated that replacing the piperazine group with a dipyrrolidine moiety would be viable. The resulting compound (**SC04**) showed lower potency than the piperazine-based compounds (IC_50_ HIV-1 YU-2 = 70 ± 6 μM; IC_50_ HIV-1 JR-CSF = 100 ± 30 μM using a pseudotyped virus in an SRIA) but demonstrated specificity, making it perfect for further optimization in potency (Figure 12).

Using Spark (Cresset, UK) and a fragment library generated from PubChem by fragmenting compounds with similarities to BMS-488043, we researched whether the dipyrrolidine could support nanomolar potency, whilst retaining specificity. Sequentially, we generated **SC07** (IC_50_ HIV-1 JR-CSF = 0.98 ± 0.06 μM in an SRIA), **SC08** (IC_50_ HIV-1 JR-CSF = 0.09 ± 0.01 μM in an SRIA), and **SC11** (IC_50_ HIV-1 JR-CSF values of 0.0008 ± 0.0004 μM in an SRIA). This was the first time that the piperazine core in this class of inhibitors had been successfully changed whilst retaining high potency. Finally, the replacement of the terminal phenyl in **SC11** by cyclohexene in **SC26** resulted in the first dipyrolrolodine-scaffolded highly potent HIV-1 entry inhibitor (IC_50_ HIV-1 JR-CSF of 2.0 ± 0.1 nM; IC_50_ HIV-1 HxBc2 of 0.6 ± 0.01 nM in an SRIA), with significantly improved predicted ADME properties (as indicated by an increase in the Oral Non-CNS Drug-like Score in StarDrop, Optibrium, UK; Figure 13).

Following this initial success, we then used similar methods to extend the core chemotypes in the HIV-1 entry inhibitor class. This resulted in the creation of five distinct core chemotypes from the original piperazine BMS-626529 that support specificity and nanomolar potencies (Figure 14 and Table 3) [53].

Stemming from these initial successes, we have since applied this workflow to identify new bioisosteres for the methyltriazole-azaindole moiety, as the main potency determinant of this entry inhibitor class and identified a compound (SC56), wherein the methyltriazole in SC28 was replaced with an amine-oxadiazole (Figure 15) [54]. This compound displayed a 1000-fold increased affinity for the B41 SOSIP Env, as judged by surface plasmon resonance (SPR).

Finally, in a soon to be published study, we have also demonstrated that SC28, with the azabicyclo-hexane core scaffold, has twice the metabolic stability of BMS-626529. This approach highlights bioisosteric replacement as a powerful tool that can be utilized to rapidly diversify chemotypes but can improve drug–target kinetic profiles and metabolism in the drug development process.

## 5. Conclusions

In this review, we demonstrated, using a few selected examples, the advantages and progress that have been made using bioisosteric replacement strategies to create new, more potent therapies to combat the AIDS pandemic. In the absence of a viable cure or vaccine, the demand for drugs targeting new therapeutic targets, in addition to new, optimized drugs to old targets is paramount. As such, we believe that the use of bioisosterism will only increase in the development of such HIV therapies in the future.

## Figures and Tables

**Figure 1 pharmaceuticals-13-00036-f001:**
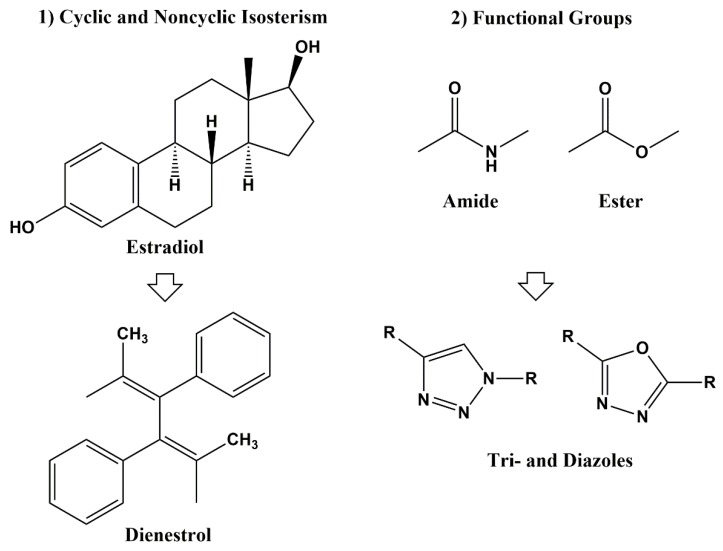
Non-Classical Bioisosterism.

**Figure 2 pharmaceuticals-13-00036-f002:**
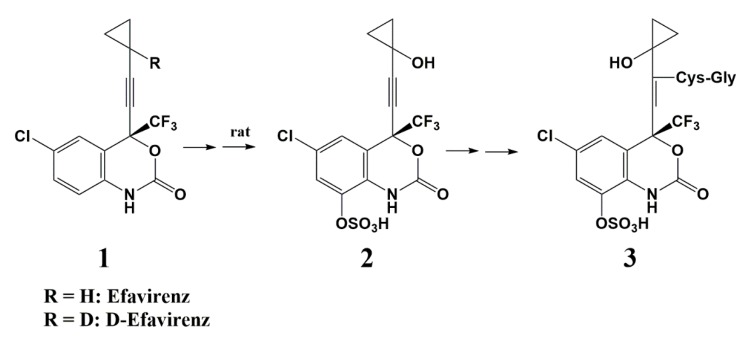
Efavirenz metabolism in rats that leads to the nephrotoxic glutathione conjugate (3).

**Figure 3 pharmaceuticals-13-00036-f003:**
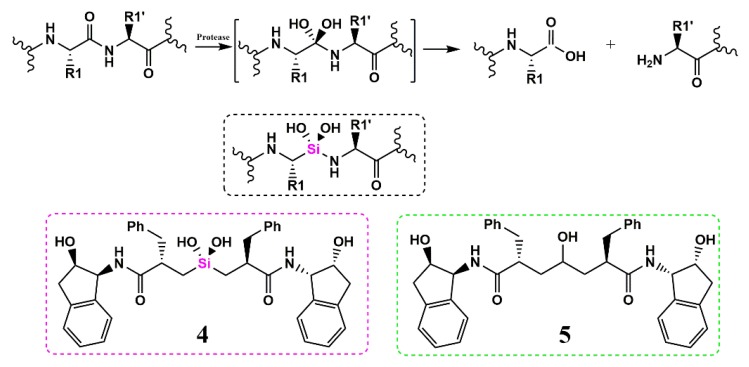
Proteolytic hydrolysis. The tetrahedral intermediate (in brackets) can be replaced by a non-hydrolyzable protease inhibitor (black dashed box).

**Figure 4 pharmaceuticals-13-00036-f004:**
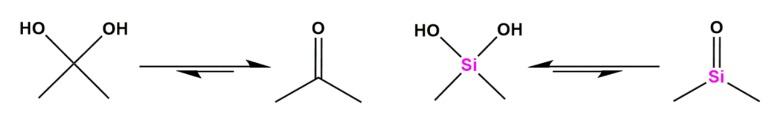
Hydration equilibrium for carbonyl (left) and silanone (right).

**Figure 5 pharmaceuticals-13-00036-f005:**
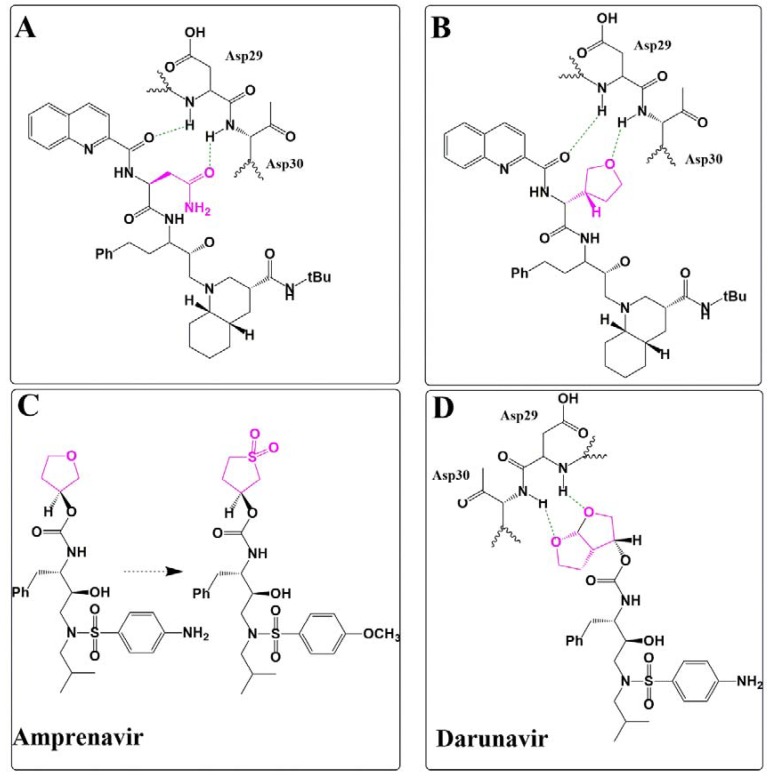
HIV-1 protease inhibitor interaction. (**A**) Saquinavir interacts with the HIV-1 protease backbone residue Asp29 and Asp30. (**B**) Asparagine moiety in (A) was replaced by 3(R)-tetrahydrofuranylglycine (in pink) to obtain Amprenavir. (**C**) A cyclic sulfone group, in which the oxygen can accept H bonds from both Asp29 and Asp30 and retain crucial backbone interactions. (**D**) Darunavir backbone interaction with the HIV-1 protease (PDB code: 2F81).

**Figure 6 pharmaceuticals-13-00036-f006:**
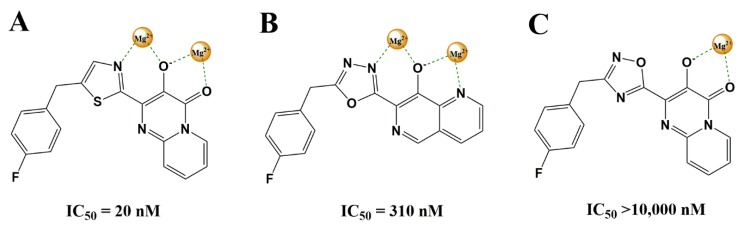
Coordination of Mg^2+^ ions by azole-substituted pyrido[1,2-a]pyrimidines (**A**) and 1,6-naphthyridines (**B**) of HIV-1 integrase. In the 1,2,4-oxadiazole (**C**) the nitrogens lone electron pair is missing, preventing coordination by a second Mg^2+^ ion. IC_50_ values were determined by using a recombinant HIV-1 integrase strand transfer assay.

**Figure 7 pharmaceuticals-13-00036-f007:**
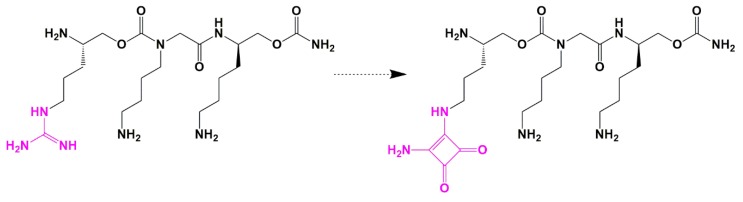
Diaminosquarate bioisostere for the guanidine moiety as a novel HIV-1 Tat-TAR RNA inhibitor.

**Figure 8 pharmaceuticals-13-00036-f008:**
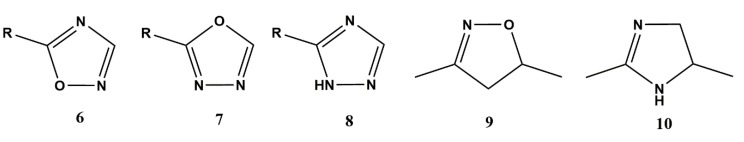
Examples of heterocyclic rings as replacements for amide and ester groups. The 1,2,3-oxadiazoles (6), 1,3,4-oxadiazoles (7), triazoles such as 1,2,4-triazoles (8), 2-isoxazoline (9), and imidazoline entities (10).

**Figure 9 pharmaceuticals-13-00036-f009:**
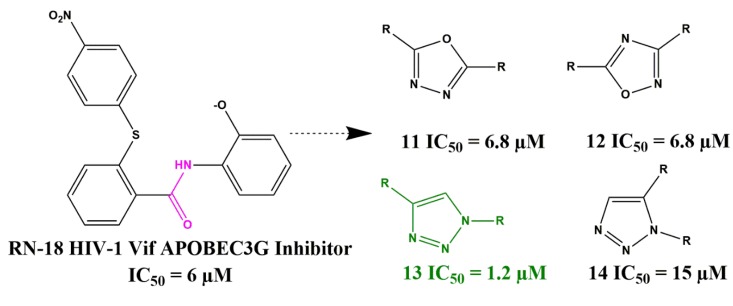
Heterocyclic bioisosteres of the RN-18 HIV-1 Vif inhibitor class and IC_50_ values obtained from a multi-round infection assay (HIV-1 LAI) using H9 and MT-4 cells.

**Figure 10 pharmaceuticals-13-00036-f010:**
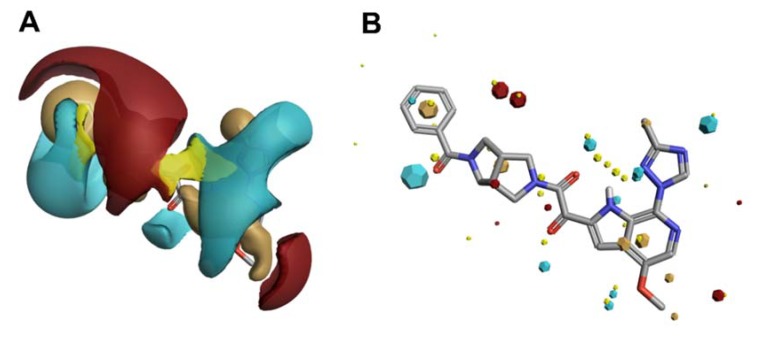
Electrostatic (blue/red), van der Waals (yellow) and hydrophobic (beige) contours (**A**) and Field Point maxima (**B**) of an HIV-1 entry inhibitor.

**Figure 11 pharmaceuticals-13-00036-f011:**
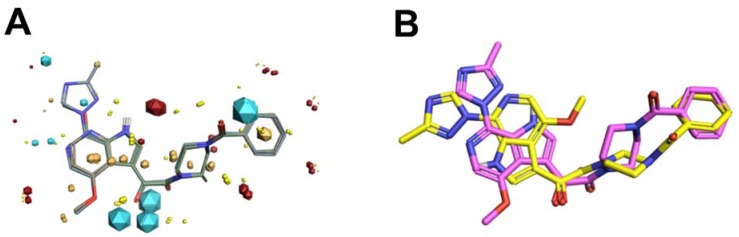
(**A**) Field-based template containing a single conformation of compounds BMS-378806 (pink), BMS-488043 (lime green), and BMS-626529 (teal green) aligned based on their three-dimensional field point patterns. Negatively charged field points are shown in blue; positively charged field points are red; van der Waals/shape field points are displayed in yellow; centers of hydrophobicity are shown in orange. The dodecahedral size of the field points is proportional to the magnitude of the extrema. (**B**) Structural superimposition of BMS-62529 conformer generated with FieldTemplater (pink) and experimentally obtained crystal structure in complex with HIV-1 BG505 SOSIP.664 (yellow, PDB code: 5U7O).

**Figure 12 pharmaceuticals-13-00036-f012:**
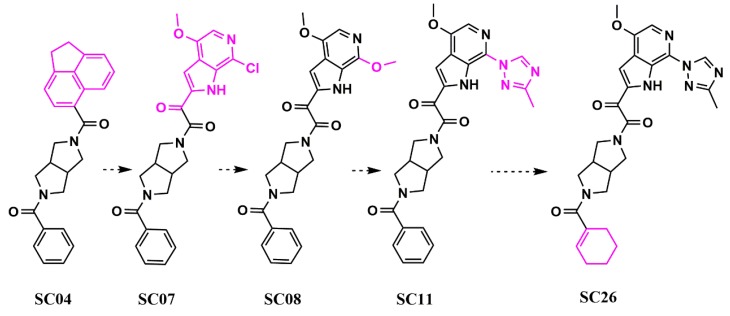
Evolution (from left to right) of HIV-1 entry inhibitors to improve potency and ADME properties. Structural changes between each compound are highlighted in pink.

**Figure 13 pharmaceuticals-13-00036-f013:**
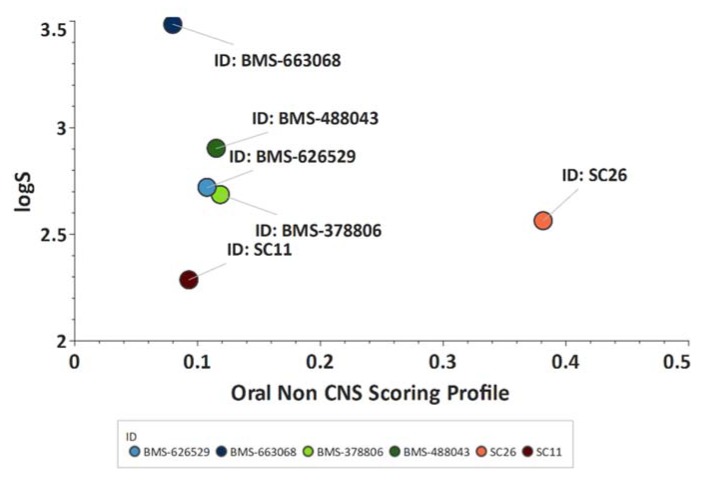
Improved ADME properties of **SC26**. Plot showing the StarDrop (Optibrium, Ltd., Cambridge, UK)–derived logS versus the score from a multimetric oral non-CNS profile for the BMS and SC compounds. Scores range from 0 to 1, with 0 suggesting extremely non-drug-like, and 1 suggesting the perfect drug.

**Figure 14 pharmaceuticals-13-00036-f014:**
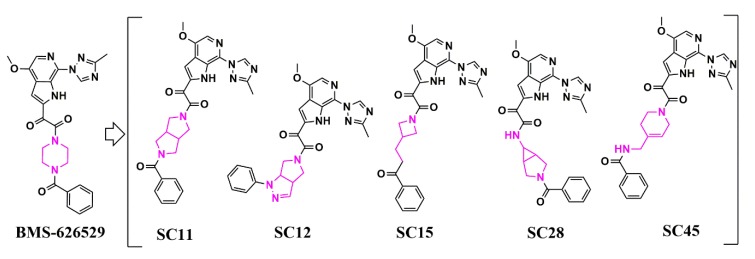
Scaffold hopping in the core region of an HIV-1 entry inhibitor. Structural changes in the core region between each compound are highlighted in pink.

**Figure 15 pharmaceuticals-13-00036-f015:**
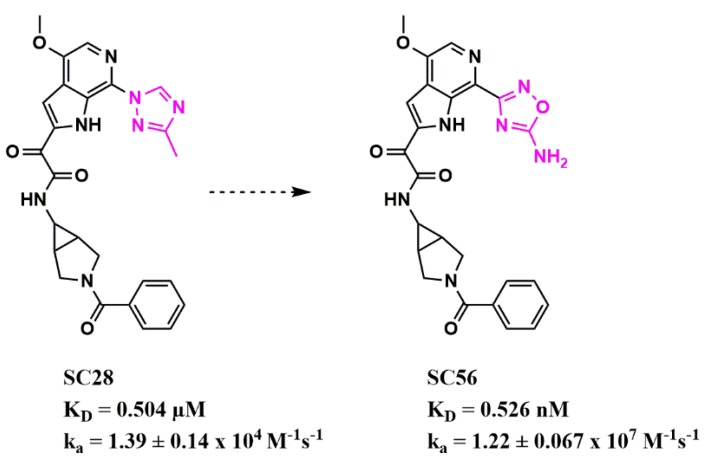
Modulation of binding kinetics using bioisosteric replacement in HIV-1 entry inhibitor design. The structural change between the two compounds is highlighted in pink. Affinity and kinetic parameters were determined using a surface plasmon resonance assay.

**Table 1 pharmaceuticals-13-00036-t001:** Grimm’s Hydride Displacement Law.

C	N	O	F	Ne	Na^+^
	CH	NH	OH	FH	-
		CH_2_	NH_2_	OH_2_	FH_2_^+^
			CH_3_	NH_3_	OH_3_^+^
				CH_4_	NH_4_^+^

**Table 2 pharmaceuticals-13-00036-t002:** Isosteres based on valence electron number.

Number of Valence Electrons				
4	5	6	7	8
N^+^	P	S	Cl	ClH
P^+^	As	Se	Br	BrH
S^+^	Sb	Te	I	IH
As^+^		PH	SH	SH_2_
Sb^+^			PH_2_	PH_3_

**Table 3 pharmaceuticals-13-00036-t003:** Specificity and potency (in µM) of SC compounds against HIV-1J R-CSF, HIV-1 B41, HIV-1 HxBc2 Env, and HIV-1 AMLV pseudotyped HIV-1 using a single-round infection assay.

Compound	IC_50_ JR-CSF	IC_50_ B41	IC_50_ HxBc2	IC_50_ AMLV
SC11	0.0008 ± 0.0001	0.002 ± 0.0002	0.001 ± 0.0001	N.A.
SC12	0.008 ± 0.002	0.006 ± 0.003	0.080 ± 0.020	N.A.
SC15	0.003 ± 0.001	0.007 ± 0.001	0.009 ± 0.001	N.A.
SC28	0.096 ± 0.019	0.085 ± 0.03	0.069 ± 0.014	N.A.
SC45	0.224 ± 0.017	0.350 ± 0.030	0.380 ± 0.030	N.A.

N.A. = not active; N.D. = not determined.

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
