# Peer review of "Bioisosteric Replacement as a Tool in Anti-HIV Drug Design"

_pharmaceuticals, 2020, doi:10.3390/ph13030036_

Round 1

Reviewer 1 Report

Dick and Cocklin review the principles of bioisosteric replacement to improve the preclinicalproperties of HIV-1 inhibitors. They provide several examples of the strategy and include real cases from the literature and from their laboratory. The review is informative and interesting and seems to be a good fit for Pharmaceuticals, but several revisions are needed to improve clarity and appeal to a wider audience:

Major points:

  1. In several cases there is limited description on the rationale for implementing bioisosteric replacement approaches to improve some classes of HIV inhibitors:
    1. For example, the paragraph in lines 120-125 is unclear – why is there a need to improve Efavirenz just because it causes renal tubular epithelial cell necrosis in rats? Some background on the justification for performing deuterium substitutions in seems necessary. Also please provide data showing improvement of compound 3 relative to 1.
    2. Similarly, for the section beginning on Line 146, if “HAART is a very effective treatment for AIDS,” then what is the need to introduce Si substitutions and to conclude in lines 166 and 167 that this opens up “future development and application for protease inhibition”? That is, if current PIs are so successful, then why do we need more?
    3. The section beginning on line 343, “inhibition of HIV-1 entry is an attractive, yet underexploited therapeutic approach, ” is confusing. There are already HIV entry inhibitors on the market (e.g., Maraviroc, Enfuvirtide, Ibalizumab), so what is the need or justification to develop new ones? What makes this “attractive” and “unexploited”?

The description of these specific examples in the Review is appropriate and warranted, but I ask that the Review provide more contexts for performing these and other studies in the first place.

  1. In almost all cases, no descriptions of the assays used to generate the reported data are provided:
    1. For example, Line 164: What assays were used to determine reported Ki values?
    2. Lines 177, 186: what assay(s) were used to determine saquinavir and derivative IC50s along with bioavailability (line 190)?
    3. Similar lack of assay descriptions are present in lines 234-5, 261, Figure 8, 384, and 382-8

  1. Several Figures are difficult to read because the fonts are small, including Figures 1 and 7 and especially Figure 5; please make these more legible.

  1. Lines 118-119: it’s not correct, at least as stated, that “continuous development of new RTIs is inevitable” as cART can stably suppress viremia in patients for decades with no resistance developing. I think the authors might mean to say that RTI resistance is always a risk, especially as a monotherapy, but this is not what gets conveyed.

  1. Section beginning on line 177: Regarding the shortcomings of saquinavir, can this be described in more detail? Specifically, please provide data on the reported bioavailability of saquinavir. Also what is the bioavailability of the eventual derivative darunavir, and how does this compare to amprenavir (and saquinavir)?

  1. Line 251: RNAP II is a host factor, not a viral factor

  1. Line 303: The authors state that they were successful in the use of molecular field points “to redesign a promising, but ultimately lacking HIV-1 entry inhibitor chemotype.” What does this mean? I don’t think the authors intend this, but it implies that they describe a failure to design HIV entry inhibitors.

  1. Line 383: What are the IC50s (and source assays) of the piperazine-based cores and BMS chemotypes, and how do they compare to SC04?

  1. Line 409 / Figure 13 – please list the nanomolar potencies and assays

Minor points:

  1. Looks like line 146 should start a new paragraph.

  1. Line 207, I think you mean “*where* entire functional groups”

Author Response

Dear reviewer 1,

Thank you very much for the opportunity to resubmit our revised draft of “Bioisosteric replacement as a tool in anti-HIV drug design” to pharmaceuticals. We greatly appreciate the time and effort you invested reading and comment on our manuscript. We are very grateful for your insightful comments. We addressed your valuable comments and hope that this alterations will improve the quality of our review.

Major points:

Comment 1: In several cases there is limited description on the rationale for implementing bioisosteric replacement approaches to improve some classes of HIV inhibitors:

  • For example, the paragraph in lines 120-125 is unclear – why is there a need to improve Efavirenz just because it causes renal tubular epithelial cell necrosis in rats? Some background on the justification for performing deuterium substitutions in seems necessary. Also please provide data showing improvement of compound 3relative to 1.

Response: Thank you for your valuable comment. The study by Mutlib et al. should highlight how deuteration can alter metabolic pathways with efavirenz as an HIV-1 example. Deuteration significantly reduced the formation of the cyclopropylcarbinol intermediate (2) and its excreted endproduct (3). This study did not provide antiviral data but is focused mainly on metabolic stability and metabolite identification/safety evaluation. We therefore rewrote the paragraph into:

Mutlib and colleagues showed that the widely used HIV-1 NNRTI efavirenz (1) produced renal tubular epithelial cell necrosis in rats (Figure 2). To understand the biochemical mechanisms of forming toxic, reactive intermediates and to characterize the safety of efavirenz, Mutlib et al. used a hydrogen-to-deuterium exchange approach to improve the metabolic stability. Efavirenz undergoes a complex metabolic transformation pathway that leads ultimately to the nephrotoxic glutathione conjugate (3). Deuteration at the cyclopropyl moiety significantly reduced the formation of the cyclopropylcarbinol intermediate (2) and the excretion of (3) in the urine of rats as quantified by LC/MS.

Figure 2: Efavirenz metabolism in rats that leads to the nephrotoxic glutathione conjugate (3).

New location: Page 6-7, line 120-127

  • Similarly, for the section beginning on Line 146, if “HAART is a very effective treatment for AIDS,” then what is the need to introduce Si substitutions and to conclude in lines 166 and 167 that this opens up “future development and application for protease inhibition”? That is, if current PIs are so successful, then why do we need more?

Response: Thank you also for this comment. We are agree that protease inhibitors are very effective, however, most of the HIV-1 protease inhibitors are accompanied by severe side effects. Silicon based protease inhibitor could overcome selectivity problems and reduce severe side effect. However, this needs further exploration. We therefore updated the paragraph with the following:

Most of the HIV-1 protease inhibitors are accompanied by side effects in long-term treatment. HIV protease inhibitor-induced metabolic syndromes can include dyslipi­demia, insulin-resistance, and lipodystrophy/lipoatrophy, as well as cardiovascular and cerebrovascular diseases. Therefore, safer and potentially promising protease inhibitor development is highly desirable. Silicon based protease inhibitors provided a proof-of-principal for the use of this isostere and future safety evaluation studies.

New location: 9, line 173-178

1.3 The section beginning on line 343, “inhibition of HIV-1 entry is an attractive, yet underexploited therapeutic approach,” is confusing. There are already HIV entry inhibitors on the market (e.g., Maraviroc, Enfuvirtide, Ibalizumab), so what is the need or justification to develop new ones? What makes this “attractive” and “unexploited”?

Response: Thank you very much for pointing this out. We updated the introduction and hope that this points improve clarity:

The inhibition of HIV-1 entry is an attractive, yet underexploited therapeutic approach for several reasons. First, much of the entry process occurs in water-soluble compartments easily accessible to drugs. Second, both viral and host cell components involved in HIV-1 entry have been identified and can be targeted. Third, because inhibition of virus entry prevents the host cell from becoming permanently infected, such inhibitors are potentially useful in many different modalities, including pre-exposure prophylactics, prophylactics, and microbicides (Ketas, T.J., et al.2007).

New location: Page 18, line 362-368

Comment 2: In almost all cases, no descriptions of the assays used to generate the reported data are provided:

  • For example, Line 164: What assays were used to determine reported Ki values?

Response: Thank you very much for this comment. We included a brief description of the assay used to derive the Ki values.

Using a high-pressure liquid chromatographic (HPLC) assay, to quantify the rate of HIV-1 protease inhibition based on the cleavage of a HIV-1 Gag peptide substrate, the silanediol (4) inhibited HIV-1 protease with a Ki of 2.7 nM and in a plaque assay with human peripheral blood mononuclear cells (PBMCs) an EC90 of 170 nM similar to the carbinol (5) (Ki = 0.37 nM, EC90 = 23 nM, Figure 3 green dashed box) was determined. This provided a proof-of-principal for the use of this isostere and has opened up this bioisostere class for future development and application for protease inhibition.

New location: Page 9, line 168-172

Comment 2.2: Lines 177, 186: what assay(s) were used to determine saquinavir and derivative IC50s along with bioavailability (line 190)?

Response: Thank you very much for this comment. We agree and included a brief description of the assay used and changed the section to:

The highly potent HIV-1 protease inhibitor saquinavir (IC50 = 0.23 nM in an in vitro HIV-1 Gag peptide substrate cleavage assay) suffers from poor bioavailability due to its peptidic nature and NH content. Absolute bioavailability in healthy volunteers receiving oral saquinavir 600mg 30 minutes after a meal was approximately 4%. The reason for its poor bioavailability is thought to be a combination of incomplete absorption and extensive first-pass metabolism. X-ray structures in complex with this inhibitor class showed extensive interactions with the protease backbone, in particular, an extensive hydrogen bond network was shown to be crucial for biological activity. However, as drug resistance emerged, Gosh and colleagues focused on the retention of a maximum number of contacts of inhibitor with the protein backbone to combat mutant enzymes.

A crucial asparagine moiety in saquinavir that contacts the backbone of Asp29 and Asp30 in HIV-1 protease (Figure 5 A), was replaced with 3(R)-tetrahydrofuranylglycine to improve inhibitory potency (Figure 5 B, IC50 = 0.05 nM in an in vitro HIV-1 Gag peptide substrate cleavage assay).

To potentially improve bioavailability, Gosh and colleagues replaced this 3(R)-tetrahydrofuranylglycine moiety in amprenavir (Ki = 160 nM) with a cyclic sulfone group, in which the oxygen can accept H-bonds from both Asp29 and Asp30 (Figure 5 C) with retained potency (Ki = 1.4 nM). To retain the ability to address both asparagine’s, further optimization did lead to the design of a more lipophilic bicyclic ether as a sulfone isostere, within the highly potent protease inhibitor darunavir (Presista®, Janssen Therapeutics), which has an absolute bioavailability of 37% (as compared to 4% of saquinavir)  and was licensed in the USA in 2006 (Figure 5 D).

New location: Page 10-11, line 188-211

Comment 2.3: Similar lack of assay descriptions are present in lines p234-5, 261, Figure 8, 384, and 382-8

Response to line234-235: Thank you very much for pointing this out. We agree and included the assay description in the figure 6:

Figure 6: Coordination of Mg2+ ions by azole-substituted pyrido[1,2-a]pyrimidines and 1,6-naphthyridines of HIV-1 Integrase. IC50 values were determined by using a recombinant HIV-1 integrase strand transfer assay.

New location: Page 13

Response to figure 8: Thank you very much for pointing this out. We agree and included the assay description in the new figure 9:

Figure 9: Heterocyclic bioisosteres of the RN-18 HIV-1 Vif inhibitor class and IC50 values obtained from a multi-round infection assay (HIV-1 LAI) using H9 and MT-4 cells.

New location: Page 16

Response to line 384 and 382-388: Thank you very much for pointing this out. We agree and included the assay description next to the IC50 values in this section:

From this field-point pharmacophore screen using Blaze (Cresset UK), we identified hits with piperazine-based cores similar to the BMS chemotypes but with potencies in the low micromolar range, ranking from 13 to 153 µM using a HIV-1 YU-2 Env pseudotyped virus in a single-round infection assay (SRIA). Therefore, to obtain a truly novel core scaffold, we chose to conduct bioisosteric replacement (scaffold hopping) with Spark (Cresset, UK). The results of this indicated that replacing the piperazine group with a dipyrrolidine moiety would be viable. The resulted compound (SC04) showed lower potency than the piperazine-based compounds (IC50 HIV-1 YU-2 = 70 ± 6 μM; IC50 HIV-1 JR-CSF = 100 ± 30 μM using a pseudotyped virus in an SRIA) but demonstrated specificity making it perfect for further optimization in potency (Figure 12).

Using Spark (Cresset, UK) and a fragment library generated from PubChem by fragmenting compounds with similarities to BMS-488043, we asked whether the dipyrrolidine could support nanomolar potency, whilst retaining specificity. Sequentially, we generated SC07 (IC50 HIV-1 JR-CSF = 0.98 ± 0.06 μM in an SRIA), SC08 (IC50 HIV-1 JR-CSF = 0.09 ± 0.01 μM in an SRIA), and SC11 (IC50 HIV-1 JR-CSF values of 0.0008 ± 0.0004 μM in an SRIA). This was the first time, the piperazine core in this class of inhibitors had been successfully changed whilst retaining high potency. Finally, a replacement of the terminal phenyl in SC11 by cyclohexene in SC26 resulted in the first dipyrolrolodine-scaffolded highly potent HIV-1 entry inhibitor (IC50 HIV-1 JR-CSF of 2.0 ± 0.1 nM; IC50 HIV-1 HxBc2 of 0.6 ± 0.01 nM in an SRIA), with significantly improved predicted ADME properties (as indicated by an increase in the Oral Non-CNS Drug-like Score in StarDrop, Optibrium, UK; Figure 13).

New location: Page 20, line 403-427

Comment 3: Several Figures are difficult to read because the fonts are small, including Figures 1 and 7 and especially Figure 5; please make these more legible.

Response: Thank you very much for this observation. We are absolutely agree and improved the resolution of these figures.

Figure 1 new location: Page 5

Figure 7 new location: Page 15

Figure 5 new location: Page 11

Comment 4: Lines 118-119: it’s not correct, at least as stated, that “continuous development of new RTIs is inevitable” as cART can stably suppress viremia in patients for decades with no resistance developing. I think the authors might mean to say that RTI resistance is always a risk, especially as a monotherapy, but this is not what gets conveyed.

Response: Thank you for pointing this out. Yes, we agree and therefore removed the sentence for clarity.

New location: Page 6, line 119-120

Comment 5: Section beginning on line 177: Regarding the shortcomings of saquinavir, can this be described in more detail? Specifically, please provide data on the reported bioavailability of saquinavir. Also, what is the bioavailability of the eventual derivative darunavir, and how does this compare to amprenavir (and saquinavir)?

Response: Thank you for pointing this out. The main shortcoming is the bioavalability and we included the absolute bioavalability of Saquinavir and Duranavir. To our knowledge, the absolute bioavalability of Amprenavir in humans was not established.

The highly potent HIV-1 protease inhibitor saquinavir (IC50 = 0.23 nM in an in vitro HIV-1 Gag peptide substrate cleavage assay) suffers from poor bioavailability due to its peptidic nature and NH content. Absolute bioavailability in healthy volunteers receiving oral saquinavir 600mg was approximately 4%. The reason for its poor bioavailability is thought to be a combination of incomplete absorption and extensive first-pass metabolism. X-ray structures in complex with this inhibitor class showed extensive interactions with the protease backbone, in particular, an extensive hydrogen bond network was shown to be crucial for biological activity. However, as drug resistance emerged, Gosh and colleagues focused on the retention of a maximum number of contacts of inhibitor with the protein backbone to combat mutant enzymes.

A crucial asparagine moiety in saquinavir that contacts the backbone of Asp29 and Asp30 in HIV-1 protease (Figure 5 A), was replaced with 3(R)-tetrahydrofuranylglycine to improve inhibitory potency (Figure 5 B, IC50 = 0.05 nM in an in vitro HIV-1 Gag peptide substrate cleavage assay).

To potentially improve bioavailability, Gosh and colleagues replaced this 3(R)-tetrahydrofuranylglycine moiety in amprenavir (Ki = 160 nM) with a cyclic sulfone group, in which the oxygen can accept H-bonds from both Asp29 and Asp30 (Figure 5 C) with retained potency (Ki = 1.4 nM). To retain the ability to address both asparagine’s, further optimization did lead to the design of a more lipophilic bicyclic ether as a sulfone isostere, within the highly potent protease inhibitor darunavir (Presista®, Janssen Therapeutics), which has an absolute bioavailability of 37% (as compared to 4% by saquinavir) and was licensed in the USA in 2006 (Figure 5 D) .

New location: Page 9-11

Comment 6: Line 251: RNAP II is a host factor, not a viral factor

Response: We thank you for pointing this out and agree that RNAP II is a host polymerase. We changed the sentence to: Subsequently, the hijacked host RNA polymerase II (RNAP II) initiates transcription of this proviral genome back into genomic RNA, and elongation is stimulated by the HIV-1 Tat protein.

New location: Page 14, line 260-263

Comment 7: Line 303: The authors state that they were successful in the use of molecular field points “to redesign a promising, but ultimately lacking HIV-1 entry inhibitor chemotype.” What does this mean? I don’t think the authors intend this, but it implies that they describe a failure to design HIV entry inhibitors.

Response: Thank you for pointing this out. What we tried to describe is the use of in silico methods for bioisosteric replacement/scaffold hopping to identify HIV-1 entry inhibitor chemotypes with improved drug-like properties and bioavalability compared to the BMS-derived compounds. We therefore updated the sentence to:

Of these computational methods of nonclassical bioisosteres identification, the use of molecular field points has gained increasing recognition over recent years and has been utilized by our group very successfully in our initial efforts to redesign a first in-class entry inhibitor to address non-optimal drug-like properties and bioavailability

New location: Page 17, line 318-321

Comment 8: Line 383: What are the IC50s (and source assays) of the piperazine-based cores and BMS chemotypes, and how do they compare to SC04?

Response: Thank you for this comment. For clarity we have now also included the IC50 values for the piperazine-based cores. The IC50 compare with SC04. We also included the assay description in this section:

From this field-point pharmacophore screen using Blaze (Cresset UK), we identified hits with piperazine-based cores similar to the BMS chemotypes but with potencies in the low micromolar range, ranking from 13 to 153 µM using a HIV-1 YU-2 Env pseudotyped virus in a single-round infection assay (SRIA). Therefore, to obtain a truly novel core scaffold, we chose to conduct bioisosteric replacement (scaffold hopping) with Spark (Cresset, UK). The results of this indicated that replacing the piperazine group with a dipyrrolidine moiety would be viable. The resulted compound (SC04) showed lower potency than the piperazine-based compounds (IC50 HIV-1 YU-2 = 70 ± 6 μM; IC50 HIV-1 JR-CSF = 100 ± 30 μM using a pseudotyped virus in an SRIA) but demonstrated specificity making it perfect for further optimization in potency (Figure 12).

New location: Page 20, line 392-401

Comment 9: Line 409 / Figure 13 – please list the nanomolar potencies and assays

Response: Thank you for pointing this out. We are agree that the IC50s values will improve the general understanding and flow of the manuscript. Therefore we included a new table 3 with IC50 values for SC11, 12, 15, 28 and 45 using a single round infection assay.

Table 3: Specificity and potency (in µM) of SC compounds against HIV-1J R-CSF, HIV-1 B41, HIV-1 HxBc2 Env, and HIV-1 AMLV pseudotyped HIV-1 using a single-round infection assay.

Compound

IC50 JR-CSF

IC50 B41

IC50 HxBc2

IC50 AMLV

SC11

0.0008 ± 0.0001

0.002 ± 0.0002

0.001 ± 0.0001

N.A.

SC12

0.008 ± 0.002

0.006 ± 0.003

0.080 ± 0.020

N.A.

SC15

0.003 ± 0.001

0.007 ± 0.001

0.009 ± 0.001

N.A.

SC28

0.096 ± 0.019

0.085 ± 0.03

0.069 ± 0.014

N.A.

SC45

0.224 ± 0.017

0.350 ± 0.030

0.380 ± 0.030

N.A.

N.A. = not active; N.D. = not determined.

New location: Page 23

Minor points:

Comment 1: Looks like line 146 should start a new paragraph.

Response: Indeed, this should be a new paragraph. We modified the accordingly.

New location: Page 8, line 150

Comment 2: Line 207, I think you mean “*where* entire functional groups”

Response: Thank you very much pointing this out to us. For clarification, we changed the sentence to:

According to Patani et al., nonclassical bioisosteres can be subdivided into the replacement of 1) cyclic groups by noncyclic groups and 2) replacement of functional groups with a variety of chemical moieties to retain biological activity (see also Figure 1).

New location: Page 12, line 221-224

Reviewer 2 Report

The authors describe bioisosteric replacement and scaffold hopping regarding the  development of new anti-HIV-1 therapeutics. The work is interesting and well written. It is clear the utility and potential of bioisosteric replacement in the continuing search for new and improved anti-HIV drugs. In my opinion, the paper might be considered to be accepted after minor revision.

Minor points

Page 5, Figure 1: Improve image resolution. Page 7, Figure 2: Improve image resolution. Page 9, Figure 5: Improve image resolution. Page 10, lines 205-207: The following sentence is not clear: “Nonclassical bioisosteres can be subdivided into the replacement of a 1) cyclic group by a noncyclic one or 2) were entire functional groups are replaced to retain biological activity.” Page 12, Figure 6: Improve image resolution and indicate each image with the letters, as in the previous figures. Page 13, lines 277: Define all structures in a single figure. Page 15, Figure 9: Indicate each image with the letters, as in the previous figures. Page 18, Figure 10: Indicate each image with the letters, as in the previous figures.

Author Response

Dear reviewer 2,

Thank you very much for the opportunity to resubmit our revised draft of “Bioisosteric replacement as a tool in anti-HIV drug design” to pharmaceuticals. We greatly appreciate the time and effort you invested reading and comment on our manuscript. We are very grateful for your insightful comments. We addressed your valuable comments and hope that these alterations will improve the quality of our review.

Minor points:

Comment 1: Page 5, Figure 1: Improve image resolution.

Response: Thank you very much for pointing this out. We agree and improved image resolution.

New location: Page 5

Comment 2: Page 7, Figure 2: Improve image resolution.

Response: Thank you very much for pointing this out. We agree and improved image resolution.

New location: Page 7

Comment 3: Page 9, Figure 5: Improve image resolution.

Response: Thank you very much for pointing this out. We agree and improved image resolution.

New location: Page11

Comment 4: Page 10, lines 205-207: The following sentence is not clear: “Nonclassical bioisosteres can be subdivided into the replacement of a 1) cyclic group by a noncyclic one or 2) were entire functional groups are replaced to retain biological activity.”

Response: Thank you very much for pointing this out. We agree and changed this sentence to:

According to Patani et al., nonclassical bioisosteres can be subdivided into the replacement of 1) cyclic groups by noncyclic groups and 2) replacement of functional groups with a variety of chemical moieties to retain biological activity (see also Figure 1).

New location: Page 12, line 221-224

Comment 5: Page 12, Figure 6: Improve image resolution and indicate each image with the letters, as in the previous figures.

Response: We agree and improved image resolution and included letters for each structure.

New location: Page 13

Comment 6: Page 13, lines 277: Define all structures in a single figure.

Response: We agree and defined all structures in a new figure 8.

New location: Page 15, line 297-299

Comment 7: Page 15, Figure 9: Indicate each image with the letters, as in the previous figures.

Response: We agree and defined a new figure 10 with A and B.

New location: Page 17

Comment 8: Page 18, Figure 10: Indicate each image with the letters, as in the previous figures. 

Response: We agree and defined a new figure 11 with A and B.

New location: Page 20

Round 2

Reviewer 1 Report

The recent revisions made by Dick and Cocklin have substantially improved their manuscript, and I am satisfied with the current version. This is an interesting review that is quite educational and fun to read, and I look forward to seeing it in print.